# The Use of Saliva Samples to Test for Congenital Cytomegalovirus Infection in Newborns: Examination of False-Positive Samples Associated with Donor Milk Use

**DOI:** 10.3390/ijns9030046

**Published:** 2023-08-17

**Authors:** Whitney Wunderlich, Abbey C. Sidebottom, Anna K. Schulte, Jessica Taghon, Sheila Dollard, Nelmary Hernandez-Alvarado

**Affiliations:** 1Care Delivery Research, Allina Health, Minneapolis, MN 55410, USA; abbey.sidebottom@allina.com (A.C.S.); anna.schulte@allina.com (A.K.S.); jessica.taghon@allina.com (J.T.); 2Division of Viral Diseases, National Center for Immunization and Respiratory Diseases, Centers for Disease Control and Prevention, Atlanta, GA 30333, USA; sgd5@cdc.gov; 3Division of Pediatric Infectious Diseases, Department of Pediatrics, University of Minnesota, Minneapolis, MN 55455, USA; hernande@umn.edu

**Keywords:** donor milk, cCMV, saliva, newborn screening, cytomegalovirus

## Abstract

A universal screening research study was conducted in six hospitals to identify the clinical sensitivity of polymerase chain reaction (PCR) testing on newborn dried blood spots (DBSs) versus saliva specimens for the diagnosis of congenital cytomegalovirus (cCMV). CMV DNA positive results from DBSs or saliva were confirmed with urine testing. Findings of several false-positive (FP) saliva PCR results prompted an examination of a possible association with donor milk. Documentation of the frequency of positive saliva results, including both true-positive (TP) and FP status from clinical confirmation, occurred. The frequency of donor milk use was compared for TP and FP cases. Of 22,079 participants tested between 2016 and 2022, 96 had positive saliva results, 15 were determined to be FP, 79 TP, and 2 were excluded for incomplete clinical evaluation. Newborn donor milk use was identified for 18 (19.14%) of all the positive saliva screens. Among the 15 FPs, 11 (73.33%) consumed donor milk compared to 7 of the 79 TPs (8.8%) (OR 28.29, 95% CI 7.10–112.73, *p* < 0.001). While milk bank Holder pasteurization inactivates CMV infectivity, CMV DNA may still be detectable. Due to this possible association, screening programs that undertake testing saliva for CMV DNA may benefit from documenting donor milk use as a potential increased risk for FP results.

## 1. Introduction

Congenital cytomegalovirus (cCMV) is the most common infectious cause of birth defects among newborns in the USA [1]. Approximately one out of every two hundred infants is born with cCMV infection and one in five positive infants will go on to have long-term health problems [1]. Infants with cCMV are at increased risk for hearing loss, developmental and motor delays, seizures, vision loss, microcephaly, and cerebral palsy. Hearing loss may appear in a child’s first four years of life, even if they pass their newborn hearing screen [2]. There is an increased focus on testing newborns for cCMV, with several states recommending targeted screening for those who fail their hearing screen. Legislation has been passed in several states to increase awareness and screening efforts [3,4,5,6,7,8]; however, there is still limited implementation of universal screening.

CMV DNA may be detected through saliva, blood, or urine. Saliva is considered a reliable specimen for CMV detection [8]. The typical false-positive (FP) rate from saliva screening among the positive samples is 7.5–13.3% [2,9,10], with breastmilk being a likely contributor [11]. However, little is known about how the consumption of donor milk may impact the FP results of cCMV screening. What is known is that CMV infection can be reactivated during lactation for a high proportion of seropositive mothers [12]; therefore, CMV DNA is likely to be present in the donated milk from those who are seropositive [13,14,15]. The reactivated virus is excreted into breast milk, but without any systemic signs and symptoms in the lactating individual. Viral DNA shedding in the lactating individual begins during the first week postpartum with low viral loads and reaches its peak viral load at weeks four to six [16]. Most studies that have examined CMV in breastmilk were related to the milk consumed by preterm infants rather than the milk used in the normal newborn care setting [13,16].

It is atypical for human milk banks to screen the donor or the milk for CMV since the virus is inactivated during Holder pasteurization; however, the virus DNA could still be present in detectible amounts, as found through polymerase chain reaction (PCR) testing [17]. Holder pasteurization is the process of heating milk to specific temperatures and then cooling it to room temperature to remove potentially harmful germs [18]. Donors typically provide milk that was secreted from 10 days to 14 months postpartum, with a majority of the donations occurring between three and six months [19]. It is unknown, but probable, that pooled donor milk could contribute to FP cCMV results in the context of saliva testing.

With the expansion of targeted and universal screening programs for cCMV, as well as the rise in the amount of human milk dispensed by human milk banks each year [20], we aimed to explore the donor milk’s contribution to FP CMV DNA saliva samples.

## 2. Materials and Methods

This study used data from a large-scale, prospective universal screening study in six hospitals from three health systems in central Minnesota, USA from February 2016 to September 2022. The study compared the clinical sensitivity of PCR performed on newborn dried blood spots (DBSs) versus PCR performed on saliva specimens in the context of universal newborn screening [10]. Partners on the study included The Centers for Disease Control and Prevention (CDC), the Minnesota Department of Health, the University of Minnesota (UMN), and three health systems (MHealth Fairview, Allina Health, and CentraCare). There are approximately 70,000 births annually in Minnesota, with approximately 25% of the state’s total births occurring at the six participating hospitals. Local institutional review board approval was obtained at each site.

Newborns whose postpartum stay occurred in either the normal newborn nursery (all sites) or the neonatal intensive care unit (NICU) (three sites) were eligible for enrollment. All study sites followed the same process for participant enrollment with an approved informed consent form and protocol for saliva collection. An informed consent discussion took place sometime between birth and hospital discharge (but no later than 21 days postpartum). Following parental consent, newborn saliva was collected. Of the families approached, 68% agreed to participate. Samples were obtained using a sterile Puritan™ polyester-tipped applicator with polystyrene handle (25–806 1PD). The swab was swirled several times along the newborn’s inner cheek and gumline. The study protocol indicated collecting saliva at least 30 min after a feeding whenever possible. Study staff documented all occasions when saliva collection could only occur within 30 min of a newborn feed. Saliva samples were dried in an open test tube for at least one hour prior to closure. These samples from each collection hospital were sent weekly to the UMN lab for testing. DBS samples were collected at 24 h and sent to the state health department as a part of routine newborn screening; a portion of this sample was then tested for CMV DNA at the UMN lab. The UMN lab tested both the saliva and DBS samples; the CDC also tested the DBS samples independently. All positive cCMV results were shared with the infant’s primary care provider by the state health department study staff. Infant care providers received an information sheet to assist in communicating positive screen results with families. Families were encouraged to have a clinical evaluation visit with their pediatrician within 21 days (where possible), for confirmatory testing to determine the final diagnostic status, as well as follow up with an infectious disease pediatrician to assess any possible cCMV-related conditions.

For this specific study examining the association between donor milk and saliva test results, patients were included if they were screened as part of the universal screening study between February 2016 and May 2022. All participants had saliva and blood tested for cCMV. Confirmatory urine PCR testing was undertaken as part of the clinical workup process, following positive results on either blood (DBS) or saliva. Patients without a final cCMV classification, due to missing clinical evaluation with urine screen, were excluded from the final analysis.

As part of typical clinical care, donor milk was available at all six study hospitals via certified human milk banks. However, each health system used different milk banks. Milk was stored frozen and thawed before consumption. Nursing staff documented feeding type in the electronic health record throughout the postpartum stay, including lot numbers of consumed donor milk.

Key measures included the saliva test results, final cCMV classification, and documented use of donor milk. Saliva tests were considered positive if there was CMV DNA detected by PCR. The first 4027 saliva swabs of the study were eluted with 300 µL sterile dH_2_O, following an establish protocol [2]. Due to early concerns with possible lower sensitivity with water elution, the method was changed to hydration of the dried swab in 300 µL of QuantaBio Extracta as previously described [10]. The data and justification for changing the saliva extraction method is described in a previous study [10]. Of the 94 positive saliva results, 13 saliva swabs were extracted by water. For PCR, 5 µL of the eluate was used in a reaction volume of 25 µL. Multiplex qPCR reaction with UL83 and NRAS primers and probes was carried out as previously described [21] using the LightCycler 96 PCR System (Roche). The PCR test was run with two replicates for each sample. For a sample to be considered positive, both replicates had to test positive. If only one replicate tested positive, the PCR test was repeated. If a sample was tested twice, and three out of four replicates were positive, then the sample was considered positive. Based on the final cCMV classifications, positive saliva results were classified as either True Positive (TP) or FP. TP cases were defined as positive saliva and a positive urine test during the clinical workup (regardless of DBS result). FP cases were defined as such if they had positive saliva, negative DBS, and a negative urine test during the clinical follow-up. We did not observe any cases where both saliva and DBS tests were positive, but the urine test was negative. Study personnel conducted chart reviews of the electronic health record to document how many newborns with TP or FP saliva classifications consumed donor milk during the delivery admission.

Descriptive statistics were conducted to compare donor milk use between the subset of TP and FP saliva cases, and an odds ratio was calculated. An additional analysis was conducted using CMV DNA PCR results of seven batches of frozen donor milk that came from the largest participating hospital site. For the PCR assay, the DNA of 100 µL of unfractionated breast milk was extracted on the QIAcube (Qiagen, Hilden, Germany) using the manufacture’s specifications for DNeasy Blood and Tissue kit. The same PCR reaction and parameters were used for the DNA eluded from the milk as the saliva swabs. The donor milk lot numbers were not the same lot numbers used to feed any of the infants with positive saliva CMV results from our study. This was because the remaining lot of milk had been distributed for patient use by the time the lab requested donor milk testing.

## 3. Results

In total, 22,079 newborns were enrolled and tested for cCMV in the universal screening study from February 2016 to May 2022. Of those, 96 saliva samples had CMV DNA detected by the PCR test. After clinical workup including a urine PCR test, the saliva results were determined to be TP for 79 cases and FP for 15 cases. Two cases were unable to be classified because the families declined clinical workup, including the urine screen, and thus were excluded.

Of the 94 cases with positive saliva results and complete clinical workup with urine screen, 18 received donor milk (19.15%). Among the TP saliva cases (*n* = 79), seven had donor milk (8.86%). Of the 15 FP saliva results, 11 used donor milk (73.33%). Donor milk, thus, was significantly associated with increased odds of FP saliva results (OR 28.29, 95% CI 7.10–112.73, *p* < 0.001) (Table 1).

Between the six screening hospitals, the FP cases were from three sites (two separate health systems). Site #4 had a large majority of the FP cases, with nine in total; eight of which had donor milk (88.90%). Site #3 had four FP cases, with three cases (75.00%) that used donor milk. Site #1 had two FP cases, but neither used donor milk.

An alternate explanation for the higher FP rate observed in the infants fed with donor milk, particularly those in the NICU, could be attributed to their older age during testing. In cases where the infants received both maternal and donor milk, it is possible that the concurrently consumed maternal milk had higher CMV levels due to a higher viral load in mature milk than colostrum; thereby, increasing the risk of FP results in saliva testing. A *t*-test was conducted comparing the age of infants between the TP and FP cases to ensure that the age of the infants did not influence the FP outcomes. The results indicated no significant difference (*p* = 0.87), with the average age of the FP cases being 1.47 days old and the TP cases being 1.96 days old at the time of collection. Among both groups, 12 infants (12.77%) were swabbed while receiving NICU care. The two oldest infants (16 and 10 days) were TP cases and had their saliva collected in the NICU. Only one FP case was detected in an infant from the NICU who had also consumed donor milk.

Among the positive saliva results, 15/94 (16.00%) were documented as having been collected within 30 min of newborn feeding. The proportion collected within 30 min was higher among the FP cases (5/15, 33.30%) compared to the TP cases (10/79, 12.70%).

The median viral load for the TP cases was 9,200,000 copies/mL of saliva with an interquartile range (IQR) from 109,000 to 79,200,000. In contrast, the median viral load of the FPs was 764 copies/mL of saliva with an IQR from 608 to 2540 (Figure 1). We conducted CMV PCR tests on seven samples, each from a different batch of donor milk. Upon analyzing the samples, we found CMV DNA in three of the seven batches. The viral loads detected in these three batches were lower than the false-positive cases, with 67, 174, and 205 copies of CMV DNA per milliliter. It is important to note that the donor milk lots that tested positive were not the same lots used to feed the infants who showed positive saliva CMV DNA results in our study. However, our examination of the samples indicates that pasteurized donor milk can contain CMV DNA.

## 4. Discussion

Our study found that donor milk use was associated with FP PCR CMV DNA saliva tests in the context of universal screening of newborns. While banked human donor milk is pasteurized, inactivating CMV infectivity, viral DNA was still present in our samples and contributed to positive results. This is the first study to examine the FP rate in a newborn saliva screening study associated with the use of donor milk. The rates of FP saliva samples in our study (0.06% out of all, and 16.00% among positive cases) was similar to those of other studies [2]. Ross et al. [9] examined FP cases among breastmilk using a combination of FP cases from the CHIMES [2] study and an outside national data source for maternal CMV seroprevalence and breastfeeding rates [22,23] to estimate the expected FP rates of 0.03% in White and 0.04% in Hispanic populations. They concluded breastfeeding in the first week contributes to *low* but acceptable FP rates for saliva [9].

The CHIMES study [2] tested 74,788 newborns for cCMV and had 307 positive saliva tests by PCR, 23 of which were determined to be FPs; a 0.03% rate among all samples and 7.50% rate among only the positive samples. Their study also found the viral loads among the FP cases were lower compared to the TPs, which was in congruence with our observations [2]. Although the FP cases generally had lower amounts of CMV DNA compared to the TP cases, there was still an overlap in the viral load measurements between both the groups. Therefore, increasing the cutoff point to eliminate the FP results would also remove several TP results, which would lower the test’s ability to identify positive cases (Figure 1).

A few studies tested donated milk samples for CMV prior to pooling and pasteurization [24,25]. Bacterial and viral analysis of unpasteurized individual milk from an internet sharing source (*n* = 101) compared to unpasteurized individual samples of milk donated to a milk bank (*n* = 20) found that 21% of the internet samples were CMV DNA positive versus 5% of the human milk bank samples [25]. In our sub-analysis of seven batches of unrelated frozen and thawed milk we found that 42% (three batches) of the milk bank samples were positive for CMV. Hamprecht et al. found that Holder pasteurization obliterates CMV infectivity in human milk but does not reduce the DNA viral load [26].

Initially, our study team noticed that when the FP cases were detected, they were occurring in batches. This led us to examine the cases by site location and consider if there were possible patterns. One observation was two FP saliva cases that occurred in a very similar timeframe. Since twins are frequently offered donor milk to help with feeding supplementation, we conducted chart reviews to find information on the feeding method. We discovered the positive CMV DNA cases that occurred close together were a twin dyad. The chart review showed they had consumed donor milk during their neonatal stay. Additionally, the research staff had documented these babies had been swabbed within 30 min of feeding. This dyad accounted for 13% (*n* = 2) of our FP cases, which lead to this investigation of FP cases.

Donated milk is mature milk, but it is colostrum or transition milk that the mother produces and the infant consumes in the hospital [27]. Previous studies of CMV seropositive mothers found 35–50% reactivated during the first week of lactation and over 90% had reactivated within four weeks post-partum [13,14,15,16]. The viral load in colostrum and transition milk is lower than in mature milk, as the peak viral load is reached at four to six weeks postpartum, with a gradual decline until there is no more viral shedding. Therefore, donated milk from seropositive mothers may have higher CMV DNA than colostrum or transition milk, depending on the timing of the donation.

In our sub-analysis that tested the unrelated frozen donor milk, we found the viral load of positive batches to be low relative to other studies of seropositive mothers. In one such study which tested the breast milk of very low birth weight premature infants in the same lab as our study, the peak viral load of the positive mothers was in the millions of copies/mL. The viral load from the analyzed donor milk in this study was comparable with the viral load of breastmilk during the first week of lactation [15].

One limitation of our study is that although we screened many newborns, the number of positive cases was quite low (0.4%) and thus the sample of FPs examined was small. It may have been advantageous to document the exact time the saliva sample was collected and the feeding method closest to the sample. Through retrospective chart review, we were able to verify that all but one NICU infant (94.44%) consumed donor milk the same day the sample was collected, but a closer time estimate would have provided further validation of these results. Additionally, in our sub-analysis we were not able to test the actual lot of donor milk that the FP cases came from due to the timing. This would have been of interest to assess the positivity and viral load among those batches. Other studies examining the association between donor milk and FP saliva PCR test rates should be conducted to verify our findings. Based on our finding that a higher proportion of the FP cases had been fed within the 30 min window of saliva collection, it may be valuable for future studies and screening programs testing saliva for CMV DNA to examine the time intervals of feeding and specimen collection. This could determine if there is an optimal timeframe that may lower the risk of sample contamination of donor milk and potential FP results.

## 5. Conclusions

Saliva remains the most commonly used specimen for detecting cCMV due to its high sensitivity and convenience of collection compared to urine. The use of human donor milk is increasing and a high proportion of people donating milk may also be shedding CMV DNA. While pasteurization inactivates CMV DNA and prevents infection to the newborn, the viral DNA may be detected in saliva PCR testing. Research studies, clinical programs, and/or newborn screening programs testing saliva for cCMV may benefit from documenting the use of donor milk in the postpartum period to potentially explain FP test findings. With the advancement of bedside saliva cCMV diagnostics, these results should be interpreted with caution. We recommend that positive cCMV screens still be confirmed alongside one or more specimens, preferably urine.

## Figures and Tables

**Figure 1 IJNS-09-00046-f001:**
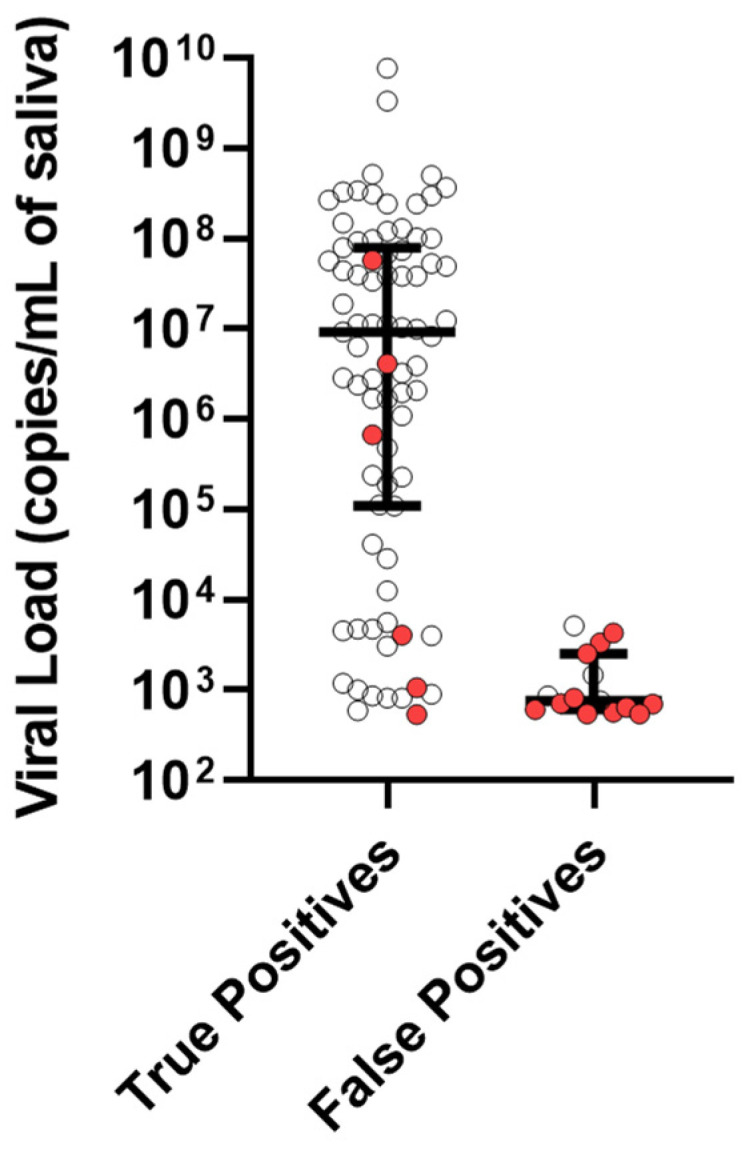
Saliva CMV viral load among true and false positive cases. Viral load in copies per mL of saliva for true positive and false positive cases. Cases with donor milk are red. The line is the median, and the error bar is the interquartile range (IQR).

**Table 1 IJNS-09-00046-t001:** Outcomes by donor milk use.

	True positive (TP)	False positive (FP)
**Donor Milk**	7	11
**No donor milk**	72	4
**Total**	79	15

## Data Availability

Data available upon request.

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
