# Peer review of "The Use of Saliva Samples to Test for Congenital Cytomegalovirus Infection in Newborns: Examination of False-Positive Samples Associated with Donor Milk Use"

_2409-515X, 2023, doi:10.3390/ijns9030046_

Round 1

Reviewer 1 Report

There were concerns about lower sensitivity with the water extraction method.  To what extent was this concern supported by the data?  If the 13 saliva swabs that were positive by water extraction were a representative sample, the remaining approximately 18,000 would have had approximately 60 additional positive results.  Your estimates for the number of positive results would then be a potential lower limit for your study population.

What was the study acceptance rate?  The participating institutions represented 25% of the birth cohort for each year.  Over 6 years this cohort would have been approximately 420,000 infants, of which 100,000 might have been eligible for the study.  Could this study acceptance rate have influenced the results?

What fraction of the 22,000 newborns who were enrolled were breast-fed or received donor milk?  As a corollary, does the inclusion of the population who were never exposed to human milk during the relevant period (through sample collection) altered your conclusions?

Reviewer 2 Report

Overall comments:

The authors take advantage of a large universal screening study to examine the contribution of banked donor breast milk to the false positive rate of saliva PCR for detection of CMV in newborns.  This is a well written manuscript and provides useful information regarding the use of saliva testing for the identification of congenital CMV.  I think their conclusion is plausible and likely correct.  However, there might be a different explanation for the seemingly higher false positive rate for infants who received donor milk and some additional details in the methods and results would be helpful to clarify.  CMV DNA PCR was detected in batches of donor milk, however the highest level (205 c/ml) was half a log lower than the median viral load in the FP saliva samples (764 c/ml).  If anything, you would expect the level in saliva to be lower than that in the breast milk due to the dilution factor.  While the authors were unable to test the actual donor milk batches provided to the infants, there is no clear reason why the fed donor milk batches should be different from the tested batches.  The premise as to why there would be a higher false positive rate with donor breast milk is that this milk is generally collected further post-partum when the CMV levels are higher, yet this was not supported by the data provided. 

An alternate explanation for the higher FP rate in donor milk-fed infants, could be that as it appears the infants fed donor milk were largely ones in the NICUs, they were older at the time of testing and if they were receiving both maternal and donor milk (as described in the scenario of the twins which prompted the study) the maternal milk they were concurrently receiving might have higher CMV levels, thus increasing the risk for FP saliva testing. 

In order to dismiss this alternate explanation, it would be helpful to know how many of the infants with FP tests were receiving both maternal and donor milk and, if they were also receiving maternal milk, the age of the infants at the time of testing and if the proximal feeding to the time of the saliva sample was with donor milk.  This latter was implied, but not specifically stated in the manuscript.

Specific comments:

Methods: 

1.       were there any infants with positive saliva, negative urine and positive DBS PCRs?  If so, were these considered TP or FP saliva tests?

2.       Was the type of feeding (maternal vs donor breast milk) most proximal to the time of saliva sampling recorded?

Results:

1.       As above, please clarify if the infants fed donor milk received exclusively donor milk or both maternal and donor milk. 

2.  If the infants received both maternal and donor breast milk, please clarify if the feeding most proximal to the saliva sampling was with maternal or donor breast milk.

3.   Clarify the age of the infants at the time of saliva testing and if there was a difference between the infants with FP vs TP saliva test results.

Reviewer 3 Report

I read with great interest your manuscript which is indeed an interesting observational study.  Unfortunately there was no ability to correlate the CMV-DNA load with the copies/mL detected in the saliva specimens.   You mentioned also in the discussion that an optimal timeframe for sampling a saliva specimen should be determined in future studies to lower the risk of sample contamination with donor milk; could a late sampling of saliva predispose towards false-negative screening results?  
The only part I didn't well understood in Materials and Methods is that you discarded the dried blood samples results in your interpretation of the results obtained in the saliva specimens: a quick search of the literature brought me across the article of Schleiss MR et al published in the journal of Int J Neonatal Screen 2023, 9, 33 that showed that the screening algorithm based on saliva and urine specimens provided  the highest sensitivity to detect cCMV.  Perhaps you should mention this article in your references.  Also emphasize that universel screening for cCMV is better compared to targeted screening which is used by ENT specialists.

Further I don't see any objection for the publication of your paper in the Int J Neonatal Screen journal.

Round 2

Reviewer 2 Report

No further comments